# Effects of producer service industry co-agglomeration and manufacturing on the green and high-end transformation of manufacturing: An empirical study from Hunan Province

**Yuchen Cheng[1], Lijun Pan[2]\*, Haobo Tan[2]\*, Ximei Liu[2]**

**1** School of Business, Xiangtan University, Xiangtan, China, **2** School of Management, Hunan Institute of Engineering, Xiangtan, China

\* yc37326@163.com (LP); 1319368928@qq.com (HT)

**Data Availability Statement:** All relevant data are within the manuscript and its Supporting Information files.

## Abstract

In the context of the "dual carbon goals" and intensified international manufacturing competition, the green and high-end transformation of manufacturing is the direction for the industry's future growth in China. The study discusses the effect of producer service industry co-agglomeration and manufacturing on the transformation of manufacturing into being green and high-end. Firstly, we systematically elaborate on the mechanism of the collaborative promotion of high-end manufacturing by the service and manufacturing industries and propose research hypotheses. Based on the 2010 to 2020 Hunan Provincial Statistical Yearbook data, we used the coupling coordination model and entropy method to calculate the level of collaborative development between the manufacturing and service industry, as well as the level of green high-end development in the manufacturing industry. Lastly, the specific impact of the synergistic effect of the two industries on the green high-end transformation of the manufacturing industry was analyzed using the dynamic panel regression model. Results found that service industry manufacturing synergy has a noteworthy positive driving effect on the green and high-end transformation of manufacturing. However, the impact varies across different service industries and manufacturing sectors with different technological levels. We also provide some implications for improving transformation efficiency in the green and high-end manufacturing industry.

## 1 Introduction

Energy efficiency emerges as a pivotal consideration for sustainable development at the national or regional level. Although the historical annual growth rate in energy efficiency has hovered around 1%, the past decade has witnessed a notable acceleration. The Chinese economy is undergoing a critical phase marked by quality, efficiency, and motivational shifts. Navigating paths for high-quality and green, low-carbon development has assumed paramount

**Funding:** The research is funded by Hunan Provincial Social Science Achievement Evaluation Committee.(No: XSP21YBZ049), received by Ximei Liu.

**Competing interests:** The authors have declared that no competing interests exist.

importance for China's continued progress. Despite the commitment to carbon peak and neutrality at the 75th United Nations General Assembly, China stands as the world's leading carbon emitter, with urban emissions contributing to about 70% of the nation's total carbon emissions [1]. Amidst a phase of high-quality economic growth, the economic model is transitioning from a "single driver" of manufacturing to a "dual-wheel drive" involving both manufacturing and service industries. The coordinated development of these sectors becomes a driving force for achieving high-quality economic development. Examining the impact of collaboration between service and manufacturing industries on the green and high-end transformation of manufacturing is crucial for the effective implementation of strategies aimed at building a robust manufacturing sector nationally and provincially. It holds significance in propelling the manufacturing industry towards higher-value production and facilitating the transition to green practices.

Existing research in this realm falls into two main categories. Firstly, studies have measured the synergy effect between service and manufacturing industries using methods such as input-output analysis, industrial synergy index, grey GM (1, N) model, and coupled coordination model [2–5]. Secondly, research has delved into the results of cooperative development between productive services and manufacturing. Productive service industries, serving as intermediate input providers, offer manufacturing industries production guarantee services and contribute to reducing production costs [6]. Scholars have conducted extensive research on productive service industries since 2000, covering aspects such as agglomeration effects, industrial collaborative development, and driving factor analysis [7–10]. Due to the inherent economic connection between productive services and manufacturing, they have become focal points in the field of industrial synergy and agglomeration. Initially, scholarly attention focused on the collaborative agglomeration phenomenon [11, 12]. However, as the pace of industrial integration development has accelerated, scholars are increasingly exploring the economic and social benefits of this collaboration [13]. Productive services act as a link between manufacturing and service provision, enhancing efficiency and competitiveness [14–17]. The service-oriented development of manufacturing positively impacts performance enhancement and industrial upgrading [18]. According to [19] the emergence of new production methods promotes the "service + manufacturing" process, creating economic advantages for the manufacturing industry. The fusion of service and manufacturing can enhance manufacturing performance to some extent through intermediate service inputs [20]. Additionally, the integration of advanced information technology with manufacturing streamlines various aspects of the manufacturing process effectively and generates significant benefits [13, 21].

With the growing concern of environmental issues from various sectors of society, there has been a considerable amount of literature examining the relationship between synergistic agglomeration and the environment. [22] measured the energy efficiency of Chinese cities from 2003 to 2016 and explored the impact of collaborative agglomeration between the two industries on energy efficiency. Research has demonstrated that collaborative agglomeration can enhance energy efficiency and facilitate initiatives for energy conservation and emission reduction. [23] used the spatial Durbin model to investigate whether collaborative agglomeration between two industries can effectively reduce air pollution. The research conclusion is positive and accompanied by spatial spillover effects. [24] used China's urban Panel data, the study discovered an 'inverted U-shaped' relationship between the collaborative agglomeration of the two industries and environmental pollution. [25] categorized the collaborative agglomeration of the two industries into two types: government-led and market-driven. Based on empirical research, it was discovered that government-led collaborative agglomeration can enhance the local ecological environment by reducing pollution levels. On the other hand,

market-led collaborative agglomeration not only enhances the local ecosystem but also has a positive impact on the ecological environment of neighboring areas.

Based on the above analysis, existing research mainly focuses on the impact of service-industry manufacturing synergy on production efficiency and innovation capabilities, while there is a lack of research on the mechanisms through which this synergy drives the green and high-end transformation of manufacturing. Previous studies indicate that specialization synergy and diversification synergy [19, 26] are the two main pathways for productive services and manufacturing synergy. However, existing research primarily focuses on the impact of diversification synergy on manufacturing efficiency and innovation capabilities, with less attention given to the effects of specialization synergy on manufacturing. Moreover, the existing literature mainly examines the national and provincial levels, paying limited attention to the industry level. In this study, we will use data from 31 sub-industries of manufacturing and 5 sub-industries of productive services in Hunan Province from 2015 to 2020 to measure the level of green and high-end transformation of manufacturing and the level of synergy between the two industries. We will empirically test the relationship between these two factors using panel regression models and explore the dynamic relationship between them. The results of this study will offer valuable insights and references for advancing the green and high-end transformation of manufacturing in Hunan Province through the collaboration between the service and manufacturing industries.

This paper's main research value can be summarized as: It aligns with the "dual-carbon" targets and focuses on the sub-industries of manufacturing in Hunan Province to examine the dynamic impact of specialization synergy between the two industries on the green and high-end transformation of manufacturing. It aims to explore the industry-specific differences in the role of specialization synergy in promoting the green and high-end development of manufacturing within the context of Hunan Province's implementation of the "strong manufacturing province" strategy.

## 2 Theoretical framework and hypotheses

This study is intricately guided by a theoretically rich framework designed to illuminate the nuanced relationships among the variables under investigation. The cornerstone of this framework lies in the Resource-Based View (RBV) model, a theoretical perspective that posits the transformative impact of effective coordination between service and manufacturing sectors on the sustainability and technological advancement of the manufacturing industry [19]. According to RBV, the symbiotic collaboration between these sectors facilitates the exchange of distinctive resources and capabilities, thereby stimulating innovation and elevating overall competitiveness within the manufacturing landscape [27]. RBV is chosen as the primary theoretical lens due to its efficacy in elucidating how the amalgamation of services and manufacturing processes can cultivate a more resilient and technologically sophisticated industrial ecosystem. The theoretical underpinning of RBV allows for a comprehensive exploration of how the collaborative dynamics between these sectors contribute to the green and high-end transformation of manufacturing industries in Hunan Province.

In tandem, this study incorporates the principles of Transaction Cost Economics (TCE) to deepen our understanding of the intricate dynamics inherent in the collaborative development of service and manufacturing industries. According to [28] TCE introduces a pragmatic perspective by emphasizing the pivotal role of transaction costs in shaping the governance structures that underpin coordination efforts. Through TCE, the study acknowledges the real-world constraints and costs associated with collaboration, offering a pragmatic lens to interpret coordination outcomes [29]. The integration of RBV and TCE theories serves as a robust

foundation for the research design, data collection, and result interpretation phases. These theoretical frameworks collectively guide the systematic exploration of how the coordinated development between service and manufacturing sectors, as informed by RBV and TCE, influences the green and high-end transformation of the manufacturing industry in Hunan Province. This holistic and theoretically grounded approach ensures a rigorous and comprehensive analysis, adding depth to our understanding of the collaborative dynamics shaping regional industrial evolution.

Previous research has indicated that the synergy between the productive services sector and the manufacturing sector can effectively enhance the technological level and production efficiency of the manufacturing sector [30, 31]. For specific details on the collaborative paths and mechanism analysis, please refer to Fig 1.

The influence of service-oriented industries on the green and high-end transformation of manufacturing varies according to their unique characteristics. Specifically, different sub-industries within the service sector exhibit distinct effects on this transformation. These effects are manifested through various types of synergistic mechanisms, which are categorized into four distinct groups. The specific details and characteristics of these mechanisms are summarized in Table 1. Based on the aforementioned analysis, this study puts forward the following hypotheses:

Hypothesis 1: Synergistic development between the service and manufacturing industries positively promotes the green and high-end transformation of manufacturing.

The collaboration between service and manufacturing industries presents industry heterogeneity in promoting the green and high-end development of manufacturing. Different sub-sectors of production-oriented service industries have distinct characteristics and collaboration paths with the manufacturing industry. Specifically, there are four collaboration paths between the logistics service industry, information service industry, and manufacturing industry, where the manufacturing industry can leverage technology, products, business, and market collaboration to improve supply chain efficiency with the logistics service industry and restructure production organization forms with the information service industry. There are

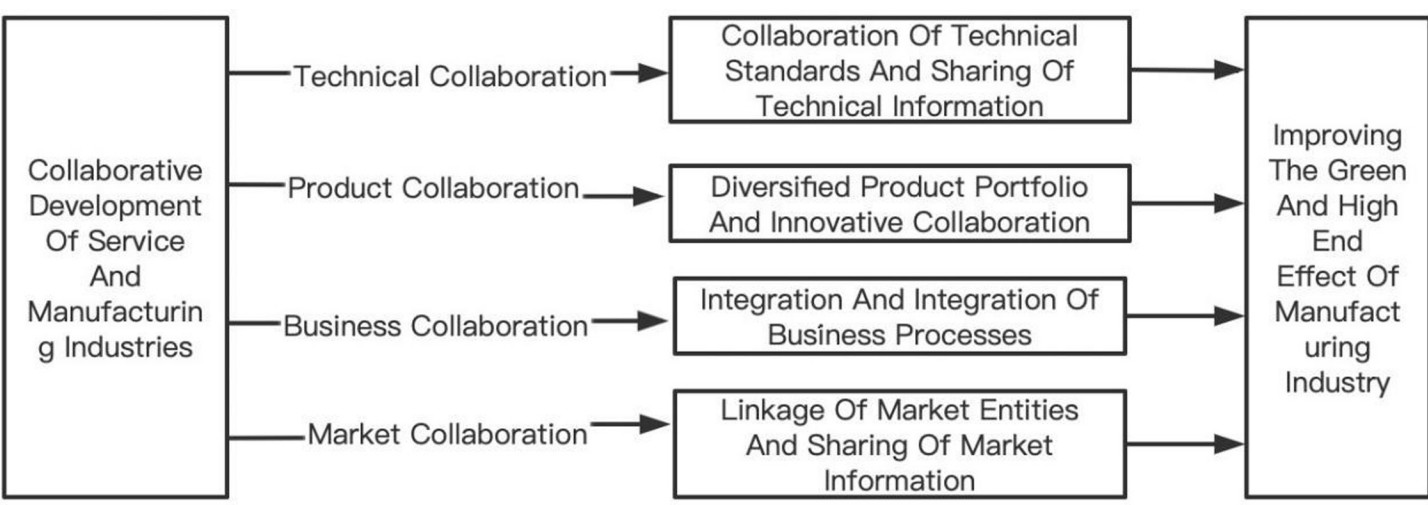

**Fig 1. The pathways of the effect of synergistic development between the service and manufacturing industries on the green and high-end transformation of manufacturing.**

Table 1. Synergistic mechanisms between various sub-sectors of producer services industry and manufacturing industry.

| Sub-sectors of the service industry. | Technological synergy. | Product synergy. | Business synergy | Market synergy |
|---|---|---|---|---|
| Logistics service industry. | Effective connection between product and raw material logistics, production technology, and facility equipment standards, as well as synergy between logistics and production data, can reduce costs and improve efficiency. | Synergizing production logistics and supply chain services can create value-added services for derived products. | Effective collaboration and integration of product, raw material logistics, and production processes can lead to cost reduction and efficiency improvement. | Two industry entities collaborate strategically to enhance the value chain. |
| Information service industry. | Sharing of data in the industrial and supply chains, intelligent analysis, and optimized decision-making can enhance the innovation capability and environmental friendliness of manufacturing | Collaborative digital technologies. Enhance the level of digitalization of products and services; Increase the value-added and environmental friendliness of products. | Industrial chain supply chain data sharing, achieving full chain data collaboration, business collaboration, and value creation collaboration, enhancing innovation capabilities, reducing costs and increasing efficiency. | Two industry entities collaborate to enhance the value chain; Engage in strategic cooperation and information sharing. |
| Financial industry | Sharing industry financial data resources can optimize investment and financing decisions in manufacturing. | Integrating financial elements, enhancing the financial attributes of products and services, and enhancing product-added value | - | Two industry entities collaborate strategically to enhance the value chain and engage in information sharing. |
| Commercial service industry | - | - | Part of management and business process outsourcing promotes business collaboration in the manufacturing industry, reduces costs, and increases efficiency. | - |
| Technology service industry | - | - | Research and development outsourcing; Drive business collaboration in the manufacturing industry; Enhance innovation capabilities and environmental friendliness. | - |

also three collaboration paths between the finance industry and the manufacturing industry, where the manufacturing industry can optimize capital and cost management through technology, products, and market collaboration. However, there is only one collaboration path between the business service industry, technology Service-based industries, and manufacturing sector, with the latter referring to the manufacturing industry that can enhance brand value through business collaboration with the former and improve R&D capabilities through business collaboration with the latter. In different levels of technological manufacturing, the collaboration between the two industries also varies. For example, low- and medium-technology manufacturing industries have limited resources to purchase technology intermediates on a large scale and face high levels of financial constraints and energy consumption. Therefore, they can share resources with the finance industry through technology collaboration. On the other hand, high-technology manufacturing industries benefit from their high-profit margins, tax incentives, and external support policies, which enable them to attract a large number of financial institutions for active interaction and integration. Thus, they can engage in strategic cooperation through technology, products, and market collaboration to enhance product value-added. Given the above, the following hypotheses are proposed in this paper:

Hypothesis 2: There are industry-specific differences in the collaboration between service and manufacturing industries for promoting the green and high-end development of manufacturing.

## 2.1 Service-industry collaboration indicators (coordination variables)

A carefully curated set of variables serves as key indicators to scrutinize the intricate landscape of coordinated development between the service and manufacturing sectors in Hunan Province. These variables are precision-engineered to measure the precise extent and nature of coordination and collaboration among specific service industries and the manufacturing sector. Focused on unraveling the impact of these coordination efforts on the green and high-end transformation of the manufacturing industry, each variable corresponds to a distinct dimension of collaboration, encompassing logistics service coordination, information service coordination, business service coordination, technology service coordination, and financial service coordination. Together, these variables form a cohesive set of pivotal indicators, shedding light on the collaborative dynamics at play and offering nuanced insights into the factors steering the sustainable and technologically advanced evolution of the manufacturing industry. They include:

- **Logistics Service Coordination:** This variable stands as a sentinel for the degree of collaboration between the logistics service industry and manufacturing. In the context of Hunan Province, the logistics sector plays a pivotal role in the efficient movement of goods, and its coordination with manufacturing is crucial for optimizing supply chain processes, reducing operational costs, and enhancing overall industrial efficiency.

- **Information Service Coordination:** Focused on evaluating the level of coordination between the information service industry and manufacturing, this variable unveils the extent to which information-based services contribute to technological advancements and operational synergy within the manufacturing sector. Information services are integral in facilitating data-driven decision-making and innovation, thereby influencing the overall transformative trajectory of manufacturing.

- **Business Service Coordination:** Measuring the extent of coordination between the business service industry and manufacturing, this variable delves into the collaborative endeavors that shape organizational management, legal consulting, and qualification investigation. Understanding the dynamics of business service coordination provides insights into the mechanisms fostering innovation, efficiency, and eco-friendly practices within manufacturing.

- **Technology Service Coordination:** Capturing the degree of coordination between the technology service industry and manufacturing, this variable sheds light on the collaborative innovation, technology transfer, and joint research projects between these sectors. In Hunan Province, technological advancements have a pivotal role in steering manufacturing toward high-end, sustainable practices, making the exploration of this coordination crucial.

- **Financial Service Coordination:** This variable gauge the level of coordination between the financial service industry and manufacturing. As the financial sector provides crucial support and investment avenues, understanding the coordination between finance and manufacturing unveils the financial mechanisms underpinning the green and high-end transformation of manufacturing industries in the region.

These variables collectively form the bedrock of this study, offering a nuanced understanding of how coordination across logistics, information, business, technology, and financial services influences the green and high-end transformation of the manufacturing sector in Hunan

Province. Each variable serves as a key facet, contributing to the holistic analysis of the collaborative dynamics between service and manufacturing industries.

# 3 Data measurement and research methods

## 3.1 Source of data collection

According to the classification of productive service industries (2019) and the classification of national economic industry. [32] defines the productive service industry as the industries within the range of G53~G60, I63~I65, L71~L72, M73~M75, J66~J69, and the manufacturing industry as the industries within the range of C13~C43. From 2015 to 2020, The Hunan Statistical Yearbook provided the source of the collected data, based on the coupling coordination values among 31 subsectors within the manufacturing sector and 5 subsectors within the productive service sector in Hunan Province from 2015 to 2020 were calculated. According to the "Classification of High-Tech Manufacturing in China (2017)," The manufacturing industry can be classified into three categories: low-tech, medium-tech, and high-tech based on their level of technological intensity [33]. The overall coupling coordination values between the three levels of manufacturing and the five sub-sectors of the producer services industry are analyzed.

## 3.2 Definitions of variables

**3.2.1 Explanatory variables.** The explanatory variable in this study is the green and high-end transformation of the manufacturing industry (gre_he). The green and high-end development of manufacturing is a dynamic process in which the manufacturing industry moves from relatively low-end and extensive development to high-end and intensive development, characterized by industrial structure upgrading, value chain climbing, green production level improvement, and competitiveness enhancement. Based on existing research results and considering data availability, this paper mainly quantifies the level of green and high-end development of manufacturing in five Dimensions of Hunan Province: industrial scale, industrial efficiency, industrial growth, innovation capability, and green level [34]. The measurement indicators specified can be seen in Table 2. The specific calculation method is to first normalize the data of each specific indicator, use the mean difference method to impute individual lost data, and then use the entropy method to calculate the weight of each indicator. Finally, using the weighted average method to calculate the level of green and high-end development of manufacturing in 31 sub-industries in Hunan Province from 2015 to 2020.

**Table 2. Evaluation indicators for green and high-end manufacturing in the manufacturing industry.**

| Evaluation Dimensions | Specific Indicators | Unit | Indicator Attribute |
|---|---|---|---|
| Industrial Scale | Output Scale | Hundred million yuan | + |
| | Employment Scale | Per 10,000 people | + |
| Industrial Efficiency | Labor Productivity of the Industry | Ten thousand per person | + |
| | Operating Revenue Profit Margin | % | + |
| Industrial Growth | Employment Growth Rate | % | + |
| | Operating Revenue Growth Rate | % | + |
| Innovation Capability | Sales Proportion of New Products | % | + |
| | Amount of Patent Applications | Pieces | + |
| Green Level | Carbon Emissions per Unit of Operating Revenue | ton | - |
| | Electricity Consumption per Unit of GDP | GWh | - |

**3.2.2 Independent variables.** The levels of coordinated development between the manufacturing industry and the logistics service industry are the independent variables in this study, information service industry, business service industry, technology service industry, and financial industry(log_con, inf_con,com_con,stu_con,fin_con). Drawing on the approach of previous studies, a model of coupling coordination degree is established to evaluate the level of coordinated development between the manufacturing industry and five sub-sectors within the service industry. A higher degree of coupling coordination indicates a higher level of integration between the two sectors. As proposed by [35, 36] the specific calculation approach is outlined as follows:

$$f(M) = \sum_{j=1}^{n} w_j^c x_{ij}^m \tag{1}$$

$$f(S) = \sum_{j=1}^{n} w_j^s x_{ij}^s \tag{2}$$

Let f(M) represent the holistic development level of the manufacturing industry subsystem, and f(S) be the comprehensive development level of a certain type of productive service subsystem, and let $w_j^c$ and $w_j^s$ be the weight of the measured indicators in the respective sub-systems of the manufacturing industry and productive service industry, which can be calculated using the entropy method [37]. Moreover, let $x_{ij}^c$ and $x_{ij}^s$ be the standardized values of the j-th indicator in the i-th year of each sub-system in the manufacturing industry and productive service industry, The coupling degree C of the system can be obtained according to Eq (3), and the system's coupling coordination degree D can be calculated according to Eq (4).

$$C = \frac{2\sqrt{f(M)*f(S)}}{f(M) + f(S)} \tag{3}$$

$$D = \sqrt{C*T}, T = af(M) + bf(S) \tag{4}$$

In the Eqs (3) and (4), the value range of C and D is [0,1]. T represents the comprehensive evaluation index of the overall synergy effect between sub-industries in the manufacturing industry and productive service industry. a and b represent their respective importance levels, and in this paper, the importance levels of both are set to be equal, so a = 0.5 and b = 0.5. In the

**Table 3. Evaluation indicators for the synergy development level between manufacturing industry and productive service industry.**

| Overall Level | Target Level | Indicator Level | Unit | Indicator Attribute |
|---|---|---|---|---|
| Synergy Development Level between Manufacturing Industry and Productive Service Industry | Industry Scale | Number of Legal Entities | unit | + |
| | | Total Assets | 1 billion yuan | + |
| | Economic Benefits | Annual Average Salary of Employees | per ten thousand people | + |
| | | Operating Revenue | 1 billion yuan | + |
| | Social Contributions | Employment | ten thousand people | + |
| | | Business Tax and Surcharge | 1 billion yuan | + |
| | industrial potential | Proportion of Fixed Asset Investment | % | + |
| | | Employment Growth Rate | % | + |

selection of synergy indicators between the two industries, the paper draws on the research of [38]. Considering the scientific validity and availability of the data, we have developed an indicator system to measure the level of synergy development between the manufacturing industry and the productive service industry, focusing on four key aspects: industry scale, economic benefits, social contributions, and industrial potential, as shown in Table 3.

**3.2.3 Control variables.** *Proportion of foreign investment($K_{MOP}$)*. The utilization level and openness of foreign investment in various sectors of the manufacturing industry can be reflected by the proportion of foreign capital. The calculation method is as follows: Foreign capital in the manufacturing industry divided by the total paid-up capital in the manufacturing industry. Foreign capital has spillover effects on manufacturing efficiency [39]. A higher proportion indicates a more open industry, which is conducive to improving manufacturing productivity and thereby facilitating the advancement of the green and high-end transformation of the manufacturing industry.

*Proportion of industrial electricity consumption($K_{ELE}$)*. Share of electricity consumption in the industrial sector reflects the production scale of the industry. According to the study [40] there is a positive correlation between electricity consumption in the manufacturing industry and production scale.

*Proportion of patent applications($K_{PAT}$)*. The innovation level of various sub-sectors within the manufacturing industry can be reflected by the proportion of patent applications. [41–43] posited that patent investment serves as a source of innovation activities, and a higher level of investment is advantageous for improving efficiency and innovation in the manufacturing industry.

## 3.3 Construction of statistical model

In this paper, panel data analysis with fixed effects and random effects models is utilized to investigate how the specialization synergy between the service industry and manufacturing industry affects the green and high-end transformation of the manufacturing sector in Hunan Province. The econometric equation is as follows:

$$gre\_he_{it} = cons_1 + \beta_{11}\ log\_con_{it} + \beta_{21}K_{it} + \lambda_{i1} + \eta_{t1} + \varepsilon_{1it} \tag{5}$$

$$gre\_he_{it} = cons_2 + \beta_{12}\ inf\_con_{it} + \beta_{22}K_{it} + \lambda_{i2} + \eta_{t2} + \varepsilon_{2it} \tag{6}$$

$$gre\_he_{it} = cons_3 + \beta_{13}\ com\_con_{it} + \beta_{23}K_{it} + \lambda_{i3} + \eta_{t3} + \varepsilon_{3it} \tag{7}$$

$$gre\_he_{it} = cons_4 + \beta_{14}stu\_con_{it} + \beta_{24}K_{it} + \lambda_{i4} + \eta_{t4} + \varepsilon_{4it} \tag{8}$$

$$gre\_he_{it} = cons_5 + \beta_{15}\ fin\_con_{it} + \beta_{25}K_{it} + \lambda_{i5} + \eta_{t5} + \varepsilon_{5it} \tag{9}$$

To ensure the reliability of the results obtained from fixed effects and random effects model regressions, a dynamic panel model approach is utilized to account for endogeneity concerns. This involves including the lagged one-period of the dependent variable in the regression equation, allowing for a more comprehensive analysis and control of potential endogenous factors.

## 4 Results and discussion

### 4.1 Regression analysis of the overall manufacturing industry

Table 4 displays the results of the fixed-effects model regression analysis conducted on panel data encompassing 31 subcategories of the manufacturing industry in Hunan Province from 2015 to 2020. The analysis reveals a statistically significant and positive relationship between specialized collaboration in the service industry and all five explanatory variables, with a significance level of 1%. This signifies that the collaborative efforts between logistics services, information services, business services, technology services, financial services, and manufacturing play a pivotal role in promoting the green and high-end transformation of the manufacturing industry in Hunan Province. Notably, the highest coefficient is associated with coordination with the business service industry, reaching 1.374. This suggests that a one-unit increase in coordination between the business service industry and manufacturing corresponds to a more than one-unit increase in the green and high-end transformation level of the manufacturing industry. This is attributed to the robust alignment between the business service industry and manufacturing in terms of industry requirements and capabilities, fostering technology spillover and input-output links, thereby facilitating organic integration, reducing pollution emissions, and enhancing the efficiency of the manufacturing industry's green and high-end transformation. Conversely, the lowest coefficient is observed for coordination with the logistics service industry, standing at 0.549, consistent with previous analyses.

Upon examining the control variables, the analysis indicates that the coefficient of the proportion of foreign investment in the manufacturing industry ($_{KMOP}$) is not significant in any of the five fixed-effects models. This aligns with [44], suggesting that the relatively low proportion of foreign capital in the actual capital of Hunan Province's manufacturing industry might explain this insignificance. Moreover, the manufacturing industry in Hunan, characterized by prominent private and state-owned enterprises, seems to diminish the significant positive influence of foreign direct investment on the shift towards green and high-end production.

**Table 4. The outcomes of the estimation for the fixed and random effects models for all subcategories of the manufacturing industry in Hunan Province.**

| Explanatory variables | fixed effects *gre_he* | | | | |
|---|---|---|---|---|---|
| *log_con* | 0.549** (0.088) | | | | |
| *inf_con* | | 0.629** (0.097) | | | |
| *com_con* | | | 1.374** (0.140) | | |
| *stu_con* | | | | 0.632** (0.092) | |
| *fin_con* | | | | | 0.977** (0.173) |
| $K_{MOP}$ | 0.168 (0.455) | 0.218 (0.452) | 0.082 (0.390) | 0.338 (0.449) | -0.526 (0.447) |
| $K_{ELE}$ | 6.474** (1.827) | 6.372** (1.812) | 4.869** (1.618) | 6.292** (1.789) | 5.365** (1.893) |
| $K_{PAT}$ | 4.680** (1.575) | 4.565** (1.563) | 3.455* (1.393) | 4.451** (1.543) | 4.488** (1.610) |
| intercept | -0.236** (0.080) | -0.270** (0.081) | -0.699** (0.096) | -0.287** (0.081) | -0.449** (0.111) |
| R-squared | 0.36 | 0.37 | 0.51 | 0.39 | 0.34 |

**、 *Significant at the 1% and 5% levels, respectively

Additionally, in all five fixed-effects models, a positive and significant relationship is observed between the proportion of industrial electricity consumption ($_{KELE}$) and the outcome variable at a 1% significance level. This implies that, after accounting for industry effects, higher industrial electricity consumption positively influences the manufacturing industry's transition towards green and high-end production. Industries with elevated electricity consumption levels tend to exhibit larger scales, superior innovation capabilities, and operational efficiency advantages. Furthermore, electricity consumption, compared to fossil fuels like coal, oil, and gas, generates lower carbon emissions, contributing to higher levels of green transformation in industries with increased electricity consumption. Lastly, the coefficient of the proportion of patent applications ($_{KPAT}$) remains consistently high and significant at the 1% level across all five fixed-effects models. This underscores that, after considering industry effects, investing in innovation activities such as patent applications positively impacts the green and high-end transformation of manufacturing.

## 4.2 Regression analysis of low, medium, and high-tech manufacturing industries

Table 5 presents a comprehensive examination of the influence exerted by specialized service industries on the advanced growth of manufacturing industries at various levels, namely low, medium, and high. The coefficients of the explanatory variables pertaining to the coordination of five specialized service industries exhibit positive benefits across all stages of manufacturing, particularly in the context of low-level manufacturing. In the context of medium-level manufacturing, it is observed that all coordination degrees, except for financial services, hold considerable significance. In the realm of advanced manufacturing, the domains that exert notable influence are limited to the coordination of commercial services and financial services, exhibiting oscillations at the levels of 1% and 5%. This highlights the industry-specific characteristic of the green and high-end transition facilitated by collaboration within the service industry. The contribution of all five categories of specialized service sectors to high-end transformation is substantial in the context of low-level manufacturing. In the realm of intermediate manufacturing, it is evident that all sectors, with the exception of financial services, exert a substantial influence on the process of transformation. The coordination of commercial and financial services is crucial for the substantial impact of high-level manufacturing.

Upon analyzing the coordination degree coefficients, it is evident that logistical, information, commercial, and technological services exhibit a significant influence on medium-level manufacturing, surpassing the overall level. The financial services sector exhibits the most substantial influence on high-level manufacturing, whereas the impact of logistical services is rather minimal. The examination of control variables indicates that the fraction of foreign capital ($_{KMOP}$) does not exhibit statistical significance in any of the models. The ratio of industrial electricity consumption ($_{KELE}$) has a noteworthy and favorable correlation in manufacturing sectors categorized as low- and medium-level, indicating the presence of industry-specific influences. In the context of low-level manufacturing, an increase in electricity consumption has a favorable impact on the process of transitioning towards environmentally sustainable practices. The significance of patent applications ($_{KPAT}$) in high-level manufacturing underscores the crucial role of innovation in facilitating the transition towards environmentally sustainable and technologically advanced production processes.

## 4.3 Robustness tests

To further validate the robustness of the conclusions and considering that the coordinated growth of the two industries is a process of dynamic evolution, and the green and high-end

**Table 5. Estimation results on the effects of coordinated development within the service industry on the green and high-end transformation of low-, medium-, and high-tech manufacturing industries in Hunan Province.**

| Explanatory variables | fixed effects (gre_he) | | | |
|---|---|---|---|---|
| | overall | low-tech | medium-tech | high-tech |
| log_con | 0.549** | 0.532** | 1.056** | 0.238 |
| | (0.088) | (0.128) | (0.165) | (0.168) |
| $K_{MOP}$ | 0.168 | -0.074 | 1.235 | -0.496 |
| | (0.455) | (0.967) | (0.828) | (0.664) |
| $K_{ELE}$ | 6.474** | 20.557** | 4.890* | 9.814 |
| | (1.827) | (5.640) | (2.288) | (5.043) |
| $K_{PAT}$ | 4.680** | 9.713 | 0.972 | 5.140** |
| | (1.575) | (6.392) | (4.725) | (1.898) |
| Intercept | -0.236** | -0.211 | -0.611** | -0.294 |
| | (0.080) | (0.115) | (0.156) | (0.164) |
| R-squared | 0.36 | 0.38 | 0.66 | 0.33 |
| inf_con | 0.629** | 0.623** | 1.189** | 0.250 |
| | (0.097) | (0.139) | (0.184) | (0.187) |
| $K_{MOP}$ | 0.218 | 0.071 | 1.542 | -0.515 |
| | (0.452) | (0.956) | (0.839) | (0.664) |
| $K_{ELE}$ | 6.372** | 18.813** | 4.924* | 9.944 |
| | (1.812) | (5.539) | (2.276) | (5.052) |
| $K_{PAT}$ | 4.565** | 9.179 | 1.378 | 5.104* |
| | (1.563) | (6.303) | (4.694) | (1.902) |
| Intercept | -0.270** | -0.234* | -0.694** | -0.298 |
| | (0.081) | (0.114) | (0.162) | (0.167) |
| R-squared | 0.37 | 0.40 | 0.66 | 0.33 |
| com_con | 1.374** | 1.268** | 1.777** | 1.182** |
| | (0.140) | (0.199) | (0.325) | (0.258) |
| $K_{MOP}$ | 0.082 | 0.141 | 0.051 | -0.097 |
| | (0.390) | (0.854) | (0.865) | (0.560) |
| $K_{ELE}$ | 4.869** | 13.353* | 5.039 | 5.240 |
| | (1.618) | (5.050) | (2.485) | (4.281) |
| $K_{PAT}$ | 3.455* | 12.802* | -2.002 | 3.790* |
| | (1.393) | (5.469) | (5.242) | (1.634) |
| Intercept | -0.699** | -0.655** | -1.029** | -0.724** |
| | (0.096) | (0.141) | (0.227) | (0.171) |
| R-squared | 0.51 | 0.52 | 0.59 | 0.52 |
| stu_con | 0.632** | 0.598** | 1.133** | 0.350 |
| | (0.092) | (0.132) | (0.183) | (0.175) |
| $K_{MOP}$ | 0.338 | 0.068 | 1.238 | -0.299 |
| | (0.449) | (0.952) | (0.846) | (0.666) |
| $K_{ELE}$ | 6.292** | 19.553** | 5.415* | 8.785 |
| | (1.789) | (5.522) | (2.330) | (4.951) |
| $K_{PAT}$ | 4.451** | 9.606 | -0.562 | 5.027** |
| | (1.543) | (6.248) | (4.856) | (1.861) |
| Intercept | -0.287** | -0.248* | -0.669** | -0.343* |
| | (0.081) | (0.115) | (0.165) | (0.164) |
| R-squared | 0.39 | 0.41 | 0.64 | 0.36 |
| fin_con | 0.977** | 0.819** | 0.554 | 1.543** |
| | (0.173) | (0.243) | (0.460) | (0.285) |
| $K_{MOP}$ | -0.526 | -0.388 | -0.393 | -0.375 |
| | (0.447) | (0.995) | (1.188) | (0.516) |
| $K_{ELE}$ | 5.365** | 10.292 | 5.212 | 7.166 |
| | (1.893) | (6.364) | (3.415) | (3.884) |

(*Continued*)

**Table 5.** (Continued）

| Explanatory variables | fixed effects (gre_he) | | | |
|---|---|---|---|---|
| | overall | low-tech | medium-tech | high-tech |
| $K_{PAT}$ | 4.488** | 18.364** | 3.004 | 2.703 |
| | (1.610) | (6.373) | (7.136) | (1.581) |
| Intercept | -0.449** | -0.350* | -0.341 | -0.906** |
| | (0.111) | (0.160) | (0.305) | (0.177) |
| R-squared | 0.34 | 0.34 | 0.24 | 0.57 |

\*\*、 \*Significant at the 1% and 5% level, respectively

transformation level of the manufacturing industry may have path dependence, this study uses a dynamic panel model for estimation, The outcomes are displayed in Table 6. The study discovered that the significance of the coefficient for the lagged one-period (L.gre_he) of the green and high-end transformation level (gre_he) of the manufacturing industry is only observed in the fixed-effects model of financial service industry coordination.

The coefficient of the lagged one period (L.gre_he) of the green and high-end transformation level (gre_he) of the manufacturing industry is only observed in the fixed effects model of the financial service industry coordination, indicating that the change in the dependent variable, the green and high-end transformation level, is persistent. Industries with higher green and high-end transformation levels in the previous period continue to perform well in the next period. The five explanatory variables have positive coefficients, which aligns with the regression findings of the baseline model in the previous text, further indicating that the specialized coordination of the five service industries can help improve the manufacturing industry's level of green and high-end transformation with a positive effect. For the control variables, their coefficients are positive and consistent with the regression findings of the baseline model in the previous text.

## 4.4 Discussion

The regression analysis of the overall manufacturing industry in Hunan Province provides compelling evidence supporting the hypotheses posited earlier.

**Hypothesis 1:** The fixed-effects model reveals a statistically significant and positive relationship between specialized collaboration in the service industry and all five explanatory variables. The highest coefficient is observed for coordination with the business service industry, emphasizing its substantial impact on the green and high-end transformation of the manufacturing industry. This suggests that the alignment between business services and manufacturing, in terms of industry requirements and capabilities, leads to greater technology spillover, reduced pollution emissions, and improved efficiency, fostering green and high-end transformation.

**Hypothesis 2:** The examination of low, medium, and high-tech manufacturing industries supports the hypothesis. The coefficients of explanatory variables for coordination with specialized service industries exhibit positive benefits across all stages of manufacturing. Notably, in low-level manufacturing, all five categories of specialized service sectors contribute substantially to high-end transformation, emphasizing the industry-specific characteristic of green and high-end transition facilitated by collaboration within the service industry.

The analysis of control variables further enhances our understanding. The proportion of foreign investment (KMOP) does not exhibit significance, suggesting that foreign capital's

**Table 6. Estimation results of the dynamic panel model.**

| Explanatory variables | fixed effects (gre_he) | | | | |
|---|---|---|---|---|---|
| L.gre_he | 0.100<br>(0.084) | 0.127<br>(0.083) | 0.060<br>(0.072) | 0.036<br>(0.085) | 0.152*<br>(0.076) |
| log_con | 0.431 **<br>(0.109) | | | | |
| inf_con | | 0.535**<br>(0.127) | | | |
| com_con | | | 1.261**<br>(0.159) | | |
| stu_con | | | | 0.600**<br>(0.140) | |
| fin_con | | | | | 1.094**<br>(0.166) |
| $K_{MOP}$ | 0.198<br>(0.587) | 0.239<br>(0.583) | 0.045<br>(0.499) | 0.218<br>(0.580) | -0.129<br>(0.528) |
| $K_{ELE}$ | 8.764**<br>(2.883) | 8.384**<br>(2.870) | 7.051**<br>(2.493) | 8.623**<br>(2.855) | 6.720*<br>(2.657) |
| $K_{PAT}$ | 4.861*<br>(1.917) | 4.581*<br>(1.909) | 3.664*<br>(1.659) | 4.797*<br>(1.898) | 3.899*<br>(1.756) |
| Intercept | -0.280*<br>(0.111) | -0.331**<br>(0.115) | -0.721**<br>(0.117) | -0.364**<br>(0.119) | -0.611**<br>(0.121) |
| R-squared | 0.30 | 0.31 | 0.48 | 0.32 | 0.42 |

**、 *Significant at the 1% and 5% levels, respectively

positive influence on green and high-end production in the manufacturing industry is not substantial in Hunan Province. Conversely, industrial electricity consumption (KELE) exhibits a positive correlation with green transformation, emphasizing its role in promoting innovation capabilities and operational efficiency in the manufacturing sector. Additionally, the proportion of patent applications (KPAT) shows a consistently high and meaningful impact on green and high-end manufacturing transformation, highlighting the positive influence of innovation activities. The robustness tests using a dynamic panel model further affirm the conclusions drawn from the fixed-effects model. The persistence of the coefficient for the lagged one-period (L.gre_he) of the green and high-end transformation level indicates path dependence, suggesting that industries with higher transformation levels in the previous period continue to perform well in the next period.

The study also examined how collaborative industrial agglomeration affects the environmentally friendly transformation of manufacturing, concentrating on Hunan Province's service sector-manufacturing coordination. Now, discuss and analyze these findings. Logistics Service Coordination and Green Transformation in manufacturing are statistically significant and positively correlated. This matches our Resource-Based View (RBV)-based theoretical paradigm. This concept states that service-manufacturing collaboration boosts technology. The findings show that logistics service coordination aids industry sustainability. This study supports it by showing a statistically significant and favorable association between Information Service Coordination and Green Transformation. This supports the Resource-Based View (RBV) model and Transaction Cost Economics (TCE) in which information services and manufacturing collaborate to foster innovation. This study shows that improved information service coordination helps the manufacturing industry transition sustainably. Business Service Coordination may help Green Transformation, according to statistical analysis. Our theoretical paradigm emphasizes business services-manufacturing integration to boost

competitiveness. A rise in business services coordination improves manufacturing sector environmental sustainability. The present study shows a positive association between Technology Service Coordination and Green Transformation. According to the Resource-Based View (RBV) and Transaction Cost Economics (TCE) frameworks, technology services, and manufacturing partnerships strengthen technical advancement and sustainability. Findings imply that technical service coordination is key to the greening industry. Our investigation shows that Financial Service Coordination and Green Transformation are positively correlated. According to our Resource-Based View (RBV) theoretical paradigm, financial services-manufacturing partnerships boost manufacturing productivity and help transition to environmentally friendly practices. The findings show that financial services sector coordination improves manufacturing industry sustainability.

### 4.5 Contribution and managerial implications

The results of the hypotheses tests underscore the integral role of collaborative industrial agglomeration in promoting the green transformation of the manufacturing industry in Hunan Province. The positive relationships between various service coordination and green outcomes highlight the multifaceted impact of coordination efforts on sustainability and technological advancement within the manufacturing sector. These findings contribute substantially to the existing literature and provide actionable insights for policymakers and industry stakeholders seeking to enhance the sustainability of manufacturing practices through coordinated service efforts. The empirical outcomes of this study highlight the intrinsic significance it holds within the field of sustainable manufacturing. By conducting a comprehensive analysis of essential data points, this research highlights the significant importance of collaborative industrial agglomeration as a crucial factor in facilitating the green transformation of the manufacturing industry. The research reveals a robust and well-supported correlation between collaborative industrial agglomeration and the environmentally sustainable transformation of the manufacturing sector. Through an examination of empirical data, this analysis aims to provide a comprehensive understanding of how service coordination contributes to environmental sustainability. By clarifying the underlying mechanisms, this study seeks to eliminate any ambiguity and shed light on the significant significance of collaborative endeavors. A thorough roadmap is offered that aligns with the specific strategic imperatives produced from collaborative industrial agglomeration. The strategic alignment of this study guarantees that its contribution is not only apparent but also practical for managers, policymakers, and other relevant stakeholders. The managerial roadmap provides a comprehensive outline of the process for incorporating environmental awareness into industrial operations, with concrete insights based on the research findings. The recommendations provided are directly associated with the overarching contribution of collaborative industrial agglomeration. This study investigates the possible consequences for sustainable development in the manufacturing sector by examining the future trajectory of collaborative industrial agglomeration. By emphasizing the long-term consequences, this highlights the lasting importance of the study's contribution to the discussion surrounding the implementation of environmentally sustainable practices in the manufacturing sector.

### 5 Conclusions

The study makes a significant contribution to the field by offering innovative insights into the intricacies of coordinated development between service and manufacturing industries, specifically contextualized within the transformative dynamics of green and high-end manufacturing in Hunan Province. Our research underscores the pivotal role of specialized collaboration,

emphasizing the influential contributions of business services in driving sustainability and technological progress within the manufacturing sector. While these findings offer valuable contributions to the existing literature, it is essential to recognize certain limitations inherent in the study, such as the reliance on retrospective data and the geographical focus on Hunan Province, which may limit the generalizability of our results. To further advance the depth and breadth of knowledge in this area, future research endeavors could broaden the study's scope by incorporating more recent and diverse datasets, allowing for a more comprehensive understanding of collaborative industrial agglomeration. Building on our innovative results, we advocate for policymakers and industry stakeholders to prioritize collaborative initiatives, particularly in business services, as a strategic avenue to propel sustainability and technological sophistication in the manufacturing industry. This recommendation entails targeted investments in innovation and infrastructure to harness the full potential of coordinated development. While our study provides crucial insights, it serves as a springboard for future research endeavors to delve into additional factors influencing collaborative industrial agglomeration, shedding light on its broader implications for sustainable manufacturing practices on a global scale.

## Supporting information

**S1 Data.**
(XLSX)

## Author Contributions

**Conceptualization:** Yuchen Cheng, Lijun Pan.

**Data curation:** Haobo Tan.

**Formal analysis:** Haobo Tan.

**Funding acquisition:** Haobo Tan.

**Investigation:** Yuchen Cheng.

**Methodology:** Yuchen Cheng.

**Project administration:** Haobo Tan.

**Resources:** Yuchen Cheng.

**Software:** Haobo Tan.

**Supervision:** Lijun Pan, Ximei Liu.

**Validation:** Yuchen Cheng, Lijun Pan, Haobo Tan.

**Visualization:** Lijun Pan, Ximei Liu.

**Writing – original draft:** Yuchen Cheng.

**Writing – review & editing:** Yuchen Cheng.

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
