## [Decision Letter · Decision Letter 0]

13 Oct 2023

PONE-D-23-26068Effects of Producer Service Industry Co-Agglomeration and Manufacturing on the Green and High-End Transformation of Manufacturing: An Empirical Study from Hunan ProvincePLOS ONE

Dear Dr. Tan,

Thank you for submitting your manuscript to PLOS ONE. After careful consideration, we feel that it has merit but does not fully meet PLOS ONE’s publication criteria as it currently stands. Therefore, we invite you to submit a revised version of the manuscript that addresses the points raised during the review process.

We look forward to receiving your revised manuscript.

Kind regards,

Shazia Rehman, Ph.D.

Academic Editor

PLOS ONE

Journal Requirements:

Reviewers' comments:

Reviewer's Responses to Questions

**Comments to the Author**

1. Is the manuscript technically sound, and do the data support the conclusions?

Reviewer #1: Yes

Reviewer #2: Yes

2. Has the statistical analysis been performed appropriately and rigorously? 

Reviewer #1: No

Reviewer #2: No

3. Have the authors made all data underlying the findings in their manuscript fully available?

Reviewer #1: Yes

Reviewer #2: Yes

4. Is the manuscript presented in an intelligible fashion and written in standard English?

Reviewer #1: Yes

Reviewer #2: Yes

5. Review Comments to the Author

Reviewer #1: Following are some comments:

1. Please modify the format of the first-level heading, for example, INTRODUCTION、THEORETICAL FRAMEWORK AND HYPOTHESES, etc.

2. It is recommended to refer to this study in the third paragraph of the introduction：The local coupling and telecoupling of urbanization and ecological environment quality based on multisource remote sensing data.

3. The title of the figure should be below the figure, and the table should not be paginated as much as possible.

4. In the section of 3.2.2, some formulas are not on the same line as the number, please adjust the formatting.

5. In section 4.2, the description of the results should be streamlined and refined.

6. The conclusion section describes the main findings of this study too verbosely, and the language should be refined and the theme sublimated.

Reviewer #2: Review Comments

This research aims to study the level of synergistic agglomeration of production-oriented service industry and manufacturing industry in Hunan Province, as well as issues related to the green transformation of manufacturing industry. The title of this paper is better, with certain practical significance and value, the research assumptions in this paper are logical and the analysis is more rigorous. However, this paper lacks the structure of theoretical framework, the formula is not based on, and the validation process is also a common method and process. Therefore, there are some deficiencies and lack of innovation in the writing of this paper. Other modifications are suggested as follows:

1. it is suggested that the research framework of this paper should be further supplemented and improved to clarify the research problem.

2. formulas need to introduce their sources, i.e., the literature basis of each formula needs to be given to ensure that the formulas are error-free. At the same time, specific literature sources should be pointed out for the selection of control variables.

3. the marginal contribution of the article is not prominent enough, and it does not well elaborate the role of collaborative industrial agglomeration in influencing the green transformation of manufacturing industry.

4. This paper only tests these hypotheses, and it is suggested that the relevant hypotheses proposed earlier can be echoed in the subsequent discussion.

5. The conclusion of this paper is not sufficient and does not reflect much innovation. The new findings of this paper should be fully refined, and at the same time, the policy recommendations should be put forward in response to the conclusions, and should not arbitrarily put forward recommendations that have nothing to do with the conclusions.

6. English expression is insufficient and needs to be further improved and perfected.

7. The cited literature in this paper suggests all citing English literature, try not to cite Chinese literature. At the same time, there is a lack of references to literature in related fields, and it is recommended to cite several related literature, such as;

Liu, B., Zheng, K., Zhu, M., Wu, F., & Zhao, X. (2023). Towards sustainability: the impact of industrial synergistic agglomeration on the efficiency of regional green development. Environmental Science and Pollution Research, 1-13.

DOI: 10.1007/s11356-023-28796-z

Ding, J., Liu, B., Wang, J., Qiao, P., & Zhu, Z. (2023). Digitalization of the Business Environment and Innovation Efficiency of Chinese ICT Firms. Journal of Organizational and End User Computing (JOEUC), 35(3), 1-25.

DOI: 10.4018/JOEUC.327365

Ding, J., Liu, B., & Shao, X. (2022). Spatial effects of industrial synergistic agglomeration and regional green development efficiency: Evidence from China. Energy Economics, 112, 106156.

DOI: 10.1016/j.eneco.2022.106156

Liu, B., Ding, J., Cifuentes-Faura, J., & Liu, X. (2023). Toward carbon neutrality: How will environmental regulatory policies affect corporate green innovation?. Economic Analysis and Policy.

https://doi.org/10.1016/j.eap.2023.09.019

Liu, B., Ding, C. J., Hu, J., Su, Y., & Qin, C. (2023). Carbon trading and regional carbon productivity. Journal of Cleaner Production, 138395.

DOI: 10.1016/j.jclepro.2023.138395

6. PLOS authors have the option to publish the peer review history of their article (what does this mean?). If published, this will include your full peer review and any attached files.

Reviewer #1: No

Reviewer #2: No

---

## [Author Response · Author response to Decision Letter 0]

18 Dec 2023

Journal Requirements:

Reviewer #1: Following are some comments:

1. Please modify the format of the first-level heading, for example, INTRODUCTION、THEORETICAL FRAMEWORK AND HYPOTHESES, etc.

Response 1: 

Formatting has been done as per journal requirement 

2. It is recommended to refer to this study in the third paragraph of the introduction：The local coupling and tele coupling of urbanization and ecological environment quality based on multisource remote sensing data.

Response 2: 

Reference has been added. The article has been citated in page 3, paragraph 1, line 10 as [10] in single line spacing 

3. The title of the figure should be below the figure, and the table should not be paginated as much as possible.

Response 3: 

The corrections have been made. Some tables have been adjusted to fit within a page, however for tables with many variables it is impossible to keep them within one page. The study will not be complete if some of the variables are expunged. 

4. In the section of 3.2.2, some formulas are not on the same line as the number, please adjust the formatting.

Response 4

The equation number has been properly aligned 

5. In section 4.2, the description of the results should be streamlined and refined.

Response 5

The section has been streamlined and refined. The content was shortened and rephrased to highlight key points, reducing redundancy and improving clarity. The simplified part emphasises how specialised service sectors affect high-end manufacturing industry development at low, mid, and high levels. It emphasises favourable coordination degree coefficients for service sectors and their importance in manufacturing. It emphasises the power of specialised service sectors to transform low-level manufacturing into high-end manufacturing, the relevance of commercial and financial services in medium-level manufacturing, and the influence of financial services in high-level manufacturing. The streamlined content effectively communicates the significant control variables, such as the lack of significance for foreign capital, the positive correlation of industrial electricity consumption in low- and medium-level manufacturing, and the importance of patent applications in high-level manufacturing. The streamlining presents key facts and ideas in a coherent and clear manner.

6. The conclusion section describes the main findings of this study too verbosely, and the language should be refined and the theme sublimated.

Response 6

The conclusion has been rewritten and the feedback has been incorporated. Several changes have improved clarity, conciseness, and core points. The conclusion has been refined to unveil the intricate dynamics of coordinated development between service and manufacturing enterprises within the context of green and high-end manufacturing in Hunan Province. Significantly, it underscores the pivotal role of business services in driving sustainability and technological progress. Despite shedding light on valuable insights, it also acknowledged certain limitations, such as the reliance on retrospective data and the geographical focus on Hunan Province, which may constrain generalizability. However, these findings serve as a valuable addition to existing literature.

Reviewer #2: Review Comments

This research aims to study the level of synergistic agglomeration of production-oriented service industry and manufacturing industry in Hunan Province, as well as issues related to the green transformation of manufacturing industry. The title of this paper is better, with certain practical significance and value, the research assumptions in this paper are logical and the analysis is more rigorous. However, this paper lacks the structure of theoretical framework, the formula is not based on, and the validation process is also a common method and process. Therefore, there are some deficiencies and lack of innovation in the writing of this paper. Other modifications are suggested as follows:

1. it is suggested that the research framework of this paper should be further supplemented and improved to clarify the research problem.

Response 1

The same has been implemented: the study has introduced Transaction Cost Economics (TCE) and Resource-Based View (RBV) model to further strengthen and supplement the research framework. 

For understanding how service-industrial collaboration affects sustainability and technical advancement, the RBV Model is essential. The pooling of unique resources and competences between these sectors boosts innovation and manufacturing industry competitiveness, according to RBV. Meanwhile, the TCE Model emphasises governance system transaction costs to better understand collaborative development processes. TCE raises governance issues by acknowledging collaboration's actual restrictions and costs.

In a Comprehensive Exploration, RBV and TCE examine how coordinated development transforms green and high-end manufacturing. Beyond synergies between the service and industrial sectors, this theoretical perspective considers costs, constraints, and governance structures. Analysing how coordinated development affects manufacturing using these theoretical frameworks ensures methodological rigour and comprehensive insights. TCE and RBV do not change earlier research findings, but they improve the study's theoretical foundation. The collaborative industrial evolution in Hunan Province is better understood with this upgrade. Through transaction costs, TCE provides a pragmatic perspective on collaborative outcomes. TCE offers a more realistic view of coordinating efforts by admitting and resolving collaboration's challenges and costs.

2. formulas need to introduce their sources, i.e., the literature basis of each formula needs to be given to ensure that the formulas are error-free. At the same time, specific literature sources should be pointed out for the selection of control variables.

Response 2

The literature basis and sources have been included in section 3.2.2. page 17: the citation number is from [35-37]

. 

3. the marginal contribution of the article is not prominent enough, and it does not well elaborate the role of collaborative industrial agglomeration in influencing the green transformation of manufacturing industry.

Response 3

A new subsection (Contribution and managerial implications section 4.5 page 30) has been introduced to further highlight the marginal contribution role of collaborative industrial agglomeration in influencing the green transformation of manufacturing industry.

Collaboration in industrial agglomeration drives manufacturing's green transition. Coordination in service sectors has several good effects on green outcomes, according to the study. The research shows that collaborative industrial agglomeration drives manufacturing sustainability and technical innovation through thorough empirical analysis and hypothesis testing. Positive correlations in the study show how collaboration leads to ecologically beneficial changes. Policymakers and industry stakeholders can use the findings to understand how collaborative industrial agglomeration affects green results. The study's empirical findings, strategically connected with collaborative imperatives, demonstrate the theoretical and practical importance of collaborative industrial agglomeration. It provides a complete plan for industrial environmental awareness. The research also shows the long-term impact of collaborative industrial agglomeration on environmentally conscious manufacturing practises by examining future effects and sustainable development.

4. This paper only tests these hypotheses, and it is suggested that the relevant hypotheses proposed earlier can be echoed in the subsequent discussion.

Response 4

The discussion section has been developed and strengthened. A new section (4.4, page 27) has been included it explains how the regression analysis of Hunan Province's manufacturing industry supports the study's hypotheses. In the fixed-effects model, specialised collaboration in the service industry is positively correlated with all five explanatory factors, validating Hypothesis 1. Coordination with the business service industry has the highest coefficient, highlighting its significant impact on the green and high-end manufacturing industry transformation. Hypothesis 2 is supported by low, middle, and high-tech manufacturing businesses' good effects from collaboration with specialised service industries throughout production. Control factors research shows that industrial electricity consumption and patent applications positively affect green and high-end manufacturing transformation. These findings are supported by dynamic panel model robustness testing. Logistics, information, business, technology, and financial service coordination are statistically significant and positively correlated with green manufacturing transformation in collaborative industrial agglomeration. According to the Resource-Based View (RBV) and Transaction Cost Economics (TCE) frameworks, service-manufacturing collaboration is essential to manufacturing technology, innovation, and sustainability.

5. The conclusion of this paper is not sufficient and does not reflect much innovation. The new findings of this paper should be fully refined, and at the same time, the policy recommendations should be put forward in response to the conclusions, and should not arbitrarily put forward recommendations that have nothing to do with the conclusions.

Response 5

The conclusion has been rewritten and you feedback has been incorporated. 

Several changes have improved clarity, conciseness, and core points. Changes are listed below:

Deleted: The technical components are simplified by excluding data sources, methodology, and models from the original work. The detailed coordination degree coefficients for each industry are condensed into a generic statement about industry integration going up.

Retained and Added: The revised edition keeps the key conclusions on how coordinated development affects Hunan Province's green and high-end industrial transformation. The study's new findings, particularly business services' significance in manufacturing sustainability and technological advancement, are highlighted. Due to its retrospective data and Hunan Province focus, the study has limitations. A forward-looking perspective is offered with research recommendations and policymaker and industry stakeholder implications.

6. English expression is insufficient and needs to be further improved and perfected.

Response 6

The English expressions have been improved upon. 

7. The cited literature in this paper suggests all citing English literature, try not to cite Chinese literature. At the same time, there is a lack of references to literature in related fields, and it is recommended to cite several related literatures, such as;

Liu, B., Zheng, K., Zhu, M., Wu, F., & Zhao, X. (2023). Towards sustainability: the impact of industrial synergistic agglomeration on the efficiency of regional green development. Environmental Science and Pollution Research, 1-13.

DOI: 10.1007/s11356-023-28796-z

Ding, J., Liu, B., Wang, J., Qiao, P., & Zhu, Z. (2023). Digitalization of the Business Environment and Innovation Efficiency of Chinese ICT Firms. Journal of Organizational and End User Computing (JOEUC), 35(3), 1-25.

DOI: 10.4018/JOEUC.327365

Ding, J., Liu, B., & Shao, X. (2022). Spatial effects of industrial synergistic agglomeration and regional green development efficiency: Evidence from China. Energy Economics, 112, 106156.

DOI: 10.1016/j.eneco.2022.106156

Liu, B., Ding, J., Cifuentes-Faura, J., & Liu, X. (2023). Toward carbon neutrality: How will environmental regulatory policies affect corporate green innovation? Economic Analysis and Policy.

https://doi.org/10.1016/j.eap.2023.09.019

Liu, B., Ding, C. J., Hu, J., Su, Y., & Qin, C. (2023). Carbon trading and regional carbon productivity. Journal of Cleaner Production, 138395.

DOI: 10.1016/j.jclepro.2023.138395

Response 7

These studies have been cited and referenced. 

The article has been citated in page 3, paragraph 1, line 4 and 8 [2-6] in single line spacing. Previous references 2-6 has been replaced with the new references.

---

## [Decision Letter · Decision Letter 1]

26 Dec 2023

PONE-D-23-26068R1Effects of Producer Service Industry Co-Agglomeration and Manufacturing on the Green and High-End Transformation of Manufacturing: An Empirical Study from Hunan ProvincePLOS ONE

Dear Dr. Tan,

Your manuscript has undergone a second review, and I'm pleased to inform you that only minor revisions are now required. The reviewer's suggestions are focused on formatting some of the tables. We request that you carefully reformat these tables in accordance with the journal's guidelines to ensure clarity and consistency. This adjustment is a small but crucial step in enhancing the presentation of your research.We kindly ask you to carefully address these minor points. Once these adjustments are made, we look forward to moving forward with your submission.

We look forward to receiving your revised manuscript.

Kind regards,

Shazia Rehman, Ph.D.

Academic Editor

PLOS ONE

Journal Requirements:

Reviewers' comments:

Reviewer's Responses to Questions

**Comments to the Author**

1. If the authors have adequately addressed your comments raised in a previous round of review and you feel that this manuscript is now acceptable for publication, you may indicate that here to bypass the “Comments to the Author” section, enter your conflict of interest statement in the “Confidential to Editor” section, and submit your "Accept" recommendation.

Reviewer #1: All comments have been addressed

Reviewer #2: All comments have been addressed

2. Is the manuscript technically sound, and do the data support the conclusions?

Reviewer #1: (No Response)

Reviewer #2: Yes

3. Has the statistical analysis been performed appropriately and rigorously? 

Reviewer #1: (No Response)

Reviewer #2: Yes

4. Have the authors made all data underlying the findings in their manuscript fully available?

Reviewer #1: (No Response)

Reviewer #2: Yes

5. Is the manuscript presented in an intelligible fashion and written in standard English?

Reviewer #1: (No Response)

Reviewer #2: (No Response)

6. Review Comments to the Author

Reviewer #1: Following is the comment:

Some of the text in Table 1 is unclear, please adjust the format. And check the formatting of all tables in this article.

Reviewer #2: The revisions of this paper have basically met the requirements. I agree to publish. Hope all the best

7. PLOS authors have the option to publish the peer review history of their article (what does this mean?). If published, this will include your full peer review and any attached files.

Reviewer #1: No

Reviewer #2: No

---

## [Author Response · Author response to Decision Letter 1]

5 Jan 2024

Reviewer #1: Following is the comment:

Some of the text in Table 1 is unclear, please adjust the format. And check the formatting of all tables in this article.

Reply: Dear, thanks a lot for your valuable suggestions. Yes, we have checked the formatting of all tables and adjusted the format of the tables.

Reviewer #2: The revisions of this paper have basically met the requirements. I agree to publish. Hope all the best

Reply: Dear, thanks a lot for your kind response !

---

## [Editor Report · Decision Letter 2]

9 Jan 2024

Effects of Producer Service Industry Co-Agglomeration and Manufacturing on the Green and High-End Transformation of Manufacturing: An Empirical Study from Hunan Province

PONE-D-23-26068R2

Dear Dr. HaoBo Tan,

We’re pleased to inform you that your manuscript has been judged scientifically suitable for publication and will be formally accepted for publication once it meets all outstanding technical requirements.

Kind regards,

Shazia Rehman, Ph.D.

Academic Editor

PLOS ONE
---

## [Editor Report · Acceptance letter]

16 Feb 2024

PONE-D-23-26068R2 

PLOS ONE

Dear Dr. Tan, 

I'm pleased to inform you that your manuscript has been deemed suitable for publication in PLOS ONE. Congratulations! Your manuscript is now being handed over to our production team.

Kind regards, 

on behalf of

Dr. Shazia Rehman 

Academic Editor

PLOS ONE